# Hyaluronic Acid Modified Au@SiO_2_@Au Nanoparticles for Photothermal Therapy of Genitourinary Tumors

**DOI:** 10.3390/polym14214772

**Published:** 2022-11-07

**Authors:** Ruizhi Wang, Nan Du, Liang Jin, Wufei Chen, Zhuangxuan Ma, Tianyu Zhang, Jie Xu, Wei Zhang, Xiaolin Wang, Ming Li

**Affiliations:** 1Department of Radiology, Huadong Hospital, Fudan University, Shanghai 200040, China; 2Department of Interventional Radiology, Zhongshan Hospital, Fudan University, Shanghai 200032, China; 3Shanghai Institute of Medical Imaging, Shanghai 200032, China

**Keywords:** genitourinary system, bladder cancer, prostate cancer, photothermal therapy, hyaluronic acid, Au nanoparticles

## Abstract

Bladder cancer and prostate cancer are the most common malignant tumors of the genitourinary system. Conventional strategies still face great challenges of high recurrence rate and severe trauma. Therefore, minimally invasive photothermal therapy (PTT) has been extensively explored to address these challenges. Herein, fluorescent Au nanoparticles (NPs) were first prepared using glutathione as template, which were then capped with SiO_2_ shell to improve the biocompatibility. Next, Au nanoclusters were deposited on the NPs surface to obtain Au@SiO_2_@Au NPs for photothermal conversion. The gaps between Au nanoparticles on their surface could enhance their photothermal conversion efficiency. Finally, hyaluronic acid (HA), which targets cancer cells overexpressing CD44 receptors, was attached on the NPs surface via 1-(3-dimethylaminopropyl)-3-ethylcarbodiimide hydrochloride (EDC) chemistry to improve the accumulation of NPs in tumor tissues. Photothermal experiments showed that NPs with an average size of 37.5 nm have a high photothermal conversion efficiency (47.6%) and excellent photostability, thus exhibiting potential application as a PTT agent. The temperature of the NPs (100 μg·mL^−1^) could rapidly increase to 38.5 °C within 200 s and reach the peak of 57.6 °C with the laser power density of 1.5 W·cm^−2^ and irradiation time of 600 s. In vivo and in vitro PTT experiments showed that the NPs have high biocompatibility and excellent targeted photothermal ablation capability of cancer cells. Both bladder and prostate tumors disappeared at 15 and 18 d post-treatment with HA-Au@SiO_2_@Au NPs, respectively, and did not recur. In summary, HA-Au@SiO_2_@Au NPs can be used a powerful PTT agent for minimally invasive treatment of genitourinary tumors.

## 1. Introduction

Many malignant tumors, such as renal cell carcinoma, bladder cancer, and prostate cancer, are widespread in the genitourinary system [1,2,3]. Among them, bladder cancer is the ninth most common cancer globally, which can lead to a death rate of 85% within 2 years of diagnosis if not treated timely [4]. Bladder cancer is mainly classified into muscle-invasive bladder cancer (MIBC) and non-muscle-invasive bladder cancer (NMIBC). Even through NMIBC can be operatively removed, it possesses a high recurrence rate and can mutate into MIBC with increased heterogeneity [5]. The most common treatment strategy for bladder cancer involves radical resection with the assistance of a platinum drug [6]. This also applies for prostate cancer, the most common malignant tumor in men [7]. However, the problem of resection is a high recurrence rate due to the incomplete removal of cancer cells [8,9]. Additionally, radical resection leads to trauma, high surgical risk, and siginificant changes in urination, which is a huge burden for the patient [10]. Therefore, identifying noninvasive or minimally invasive treatment methods to remove cancer cells and preserving patient organs is warranted.

Photothermal therapy (PTT) possesses the advantages of deep tissue penetration, low side effect, and non-invasiveness, making it be widely used to treat malignant tumors of the urinary system [11,12,13]. However, conventional photosensitizers are not suitable for clinical applications due to their low selectivity and biocompatibility [14,15]. To overcome these limitations, hyaluronic acid (HA)-modified nanoparticles (NPs) with high selectivity for tumors, high therapeutic efficiency, and excellent biocompatibility have been fabricated [16,17,18,19]. Those NPs are capable of efficient diagnosis or therapy of bladder cancer cells and prostate cancer cells overexpressing CD44 receptor [20,21].

The local surface plasmon resonance effect of gold (Au) NPs makes them an excellent agent to absorb and scatter light [22]. The photothermal conversion by Au NPs elevates the surrounding temperature rapidly, enabling the applications of photothermal imaging, PTT, etc. [23]. The preparation and characterization analysis of Au@SiO_2_@Au NPs had been carried out [24,25], especially the photothermal conversion mechanism and ability [24,26]. For Au nanoclusters, their photothermal properties are determined not only by the size and morphology but also interparticle gap. For example, the deposition of Au nanoparticles on SiO_2_ surface via seed growth method leads to high photothermal conversion rate and excellent biocompatibility due to the presence of gaps between Au NPs [27]. Moreover, the fluorescence characteristics of Au NPs enable timely treatment feedback through real-time imaging.

In this study, glutathione (GSH) was used as the template to prepare fluorescent Au NPs, followed by coating of SiO_2_ via the sol-gel process. Au clusters were deposited on the surface of SiO_2_ via seed growth method. Then, HA was covalently modified onto the NP surface to construct multifunctional HA-Au@SiO_2_@Au with the capabilities of cancer cell targeting, fluorescence imaging, and photothermal conversion (Figure 1). Evaluating the photothermal properties of HA-Au@SiO_2_@Au in vitro and in vivo revealed that those NPs had an excellent photothermal therapy effect, benefiting from the targeting ability. Meanwhile, those NPs exhibited good biocompatibility and favorable photostability.

## 2. Materials and Methods

### 2.1. Reagents and Instruments

*N*-Hydroxysuccinimide (NHS), 1-(3-dimethylaminopropyl)-3-ethylcarbodiimide hydrochloride (EDC), HA, chloroauric acid (HAuCl_4_), tetraethyl orthosilicate (TEOS), 3-aminopropyl-triethoxysilane (APTES), and 5-L-glutamyl-L-cysteinylglycine (GSH) were purchased from Sigma-Aldrich (St. Louis, MO, USA). Hydrochloric acid (HCl), potassium carbonate (K_2_CO_3_), methanamide (CH_3_NO), and methyl alcohol (MeOH) were purchased from Aladdin Reagent Co., Ltd. (Shanghai, China). Fetal bovine serum (FBS), trypsin-EDTA (0.25% solution), penicillin-streptomycin solution, RPMI 1640 medium, and phosphate buffer saline (PBS) were purchased from Gibco. The Cell Counting Kit-8 (CCK-8) was purchased from Beyotime (Shanghai, China). All reagents were analytically pure. The experimental water used was ultrapure. The TEM (JEOL-2100, JEOL, Akishima, Japan) was operated at 200 kV. Zeta potential measurements were performed using a Malvern Zetasizer Nano ZS model ZEN3600 (Worcestershire, UK).

### 2.2. Synthesis of Au@SiO_2_@Au NPs

First, fluorescent Au NPs were prepared as described [28,29,30,31]. Gold NPs were prepared using glutathione as template. Au NPs were prepared by stirring HAuCl_4_ (2 mL, 20 mM) and glutathione (0.5 mL, 10 mg/mL) for 24 h and drying. Then, Au@SiO_2_@Au NPs were prepared. Briefly, the Au NP (2 mL, 18 mg/mL) solution was added dropwise with NaOH (20 μL, 0.1 mM). After stirring for 30 min, TEOS (18 μL) was added with continuous stirring for 36 h, washed, and centrifuged to obtain Au@SiO_2_. The obtained Au@SiO_2_ methanol solution (4 mL, 18 mg/mL) was added to APTES (10 μL), and the pH of the solution was adjusted to 1 with hydrochloric acid, heated to reflux, and centrifuged to obtain aminated Au@SiO_2_ NPs. Finally, Au@SiO_2_@Au was prepared using the gold seed growth method. A total of 2 mL of 0.1-M NaOH was added to 3 mL of 20-mM HAuCl_4_ and stirred for 15 min. Then, 1-mL Au@SiO_2_-NH_2_ (18 mg/mL) solution was added, and the temperature was raised to 70 °C with stirring. Washing, centrifuging, and drying were performed to obtain Au@SiO_2_@Au NPs of gold seeds. The K_2_CO_3_-HAuCl_4_ solution was further added to finally form Au@SiO_2_@Au NPs.

### 2.3. Synthesis of HA-Au@SiO_2_@Au NPs

HA-Au@SiO_2_@Au was prepared as described [32,33,34]. First, the HA-NHS ester was synthesized; NHS (5.7 mg, 50 μmol) and EDC (10.3 mg, 50 μmol) were added to the formamide solution of HA (4 mL, 5 μmol). After the mixture was stirred at 24 °C for 10 h, Au@SiO_2_@Au was added to the HA-NHS ester mixture (weight ratio of Au@SiO_2_@Au to HA, 5:1) and stirred for 10 h. After centrifugation and drying, HA-Au@SiO_2_@Au NPs were obtained.

### 2.4. Photothermal Properties of HA-Au@SiO_2_@Au NPs

The NIR laser was generated with the infrared diode laser source (Changchun New Industry Photoelectric Technology, Changchun, China) and used for all studies. For in vitro studies, a series of concentrations of Au@SiO_2_@Au and HA-Au@SiO_2_@Au NPs (12.5, 25, 50, 100, 200 and 400 μg·mL^−1^) aqueous solutions were placed in a quartz colorimetric dish and irradiated by the NIR laser (808 nm, 0.5–2.0 W·cm^−2^). Temperature changes were measured using a thermometer. Infrared thermography was used to record temperature changes (FLIR A300, FLIR System, Wilsonville, OR, USA).

### 2.5. Cell Culture

Prostate cancer cells (PC-3) and bladder cancer cells (MB49) were purchased from the Cell Bank of the Chinese Academy of Sciences (Shanghai, China). The cells were cultured in RPMI 1640 medium supplemented with 5% FBS and 1% penicillin-streptomycin solution in a humidifying incubator at 37 °C with 5% CO_2_. Cells were cultured to six generations for experiments.

### 2.6. PTT of Cancer Cells In Vitro

CCK-8 was used to evaluate cell viability. Briefly, bladder and prostate cancer cells (density: 10^6^/mL) were inoculated in 96-well plates, and different concentrations of Au@SiO_2_@Au and HA-Au@SiO_2_@Au NPs were added for 24 h. After irradiation with 808-nm laser under different conditions (concentration, frequency, time), the medium was replaced, and cells were cultured for 24 h and incubated with 1% CCK-8 for 1 h. Then, absorbance was measured using an enzyme-labeled instrument.

### 2.7. Quantitative Analysis of Cancer Cells Phagocytosis

Bladder and prostate cancer cells were inoculated into a 24-well plate and incubated for 24 h. Then, 1% of 100 μg/mL Au@SiO_2_@Au and HA-Au@SiO_2_@Au NPs were added for different incubation times. The fluorescence images were obtained using an excitation wavelength of 405 nm, followed by being analyzed by the Image J (1.46a, NIH Image J system, Bethesda, MD, USA).

### 2.8. PTT of Tumors In Vivo

BALB/c nude mice were used for the in vivo photothermal study of HA-Au@SiO_2_@Au NPs. Twelve 6-week-old male mice were subcutaneously injected with 10^7^/mL PC-3 and MB49 cells. Intravenous injection and laser irradiation were started after the tumor volume was approximately 100 mm^3^. Mice were anesthetized with isoflurane in vitro. This animal study was reviewed and approved by the Institutional Animal Care and Use Committee of Zhongshan Hospital, Fudan University.

## 3. Results and Discussion

### 3.1. Characterization

TEM characterization of the prepared HA-Au@SiO_2_@Au NPs is shown in Figure 1A. However, the TEM image is not very clear likely due to the coating of a large amount of HA on the particle surface [35]. Nevertheless, some darkened Au nanocrystals are observed in the nanocomposites. The diameter of HA-Au@SiO_2_@Au NPs was analysized to be 37.5 ± 4.5 nm. Zeta potential analysis revealed that Au@SiO_2_@Au and HA-Au@SiO_2_@Au NPs were negatively charged (−7.2 ± 3.8 and −18.8 ± 4.2 mV) (Figure 1B), indirectly proving that the NPs’ surface was modified with the modifier HA to allow for cell recognition.

### 3.2. Photothermal Properties

To study in vitro photothermal characteristics, PBS solutions of Au@SiO_2_@Au and HA-Au@SiO_2_@Au NPs were irradiated with 808-nm laser, and temperature changes were recorded. The NP concentrations used were 12.5, 25, 50, 100, 200, and 400 μg·mL^−1^ and the irradiation experiment was carried out at 1.5 W·cm^−2^ for 600 s. Simultaneously, 100 μg·mL^−1^ of NPs were irradiated with variable power settings of 0.5, 1, 1.5, and 2 W·cm^−2^ for 600 s. Additionally, 100 μg·mL^−1^ Au@SiO_2_@Au and HA-Au@SiO_2_@Au NPs were repeatedly irradiated at 1000 s intervals to verify their stability.

Photothermal conversion efficiency (*η*) was calculated using the following formula [36]:$$\eta =\frac{hS\left(T_{max}-T_{sur}\right)-Q_{s}}{I\left(I-10^{-A}\right)}$$
where *h* is the heat transfer coefficient, *S* is the surface area of the container, *T_max_* is the equilibrium temperature of the sample solution, *T_sur_* is the ambient temperature, *Q_s_* is the heat generated by the container and water under laser irradiation, *I* is the laser power, and *A* is the absorbance of the sample solution at 808 nm.

The maximum temperature of HA-Au@SiO_2_@Au NPs is positively correlated with the concentration at a given range, whereas the difference in maximum temperature was not obvious when the concentration was higher than 100 μg·mL^−1^ (Figure 2A). Furthermore, monitoring the temperature change revealed that the temperature of Au@SiO_2_@Au and HA-Au@SiO_2_@Au NPs rapidly increased by approximately 40 °C (39 °C and 38.5 °C) within 200 s and their top temperatures were 58.8 °C and 57.6 °C respectively with the irradiated time of 600 s (Figure 2B), indicating that those NPs had an excellent photothermal conversion ability. Therefore, HA-Au@SiO_2_@Au NPs exhibit a concentration-dependent photothermal conversion capacity, indicating that heat generation can be rationally regulated. Using the photothermal conversion rate formula, *η* = 47.6% was obtained. We found that the highest temperature of HA-Au@SiO_2_@Au NPs was almost unchanged during the cyclic stability test, which confirmed the good photostability (Figure 2C). Overall, the unique photothermal conversion ability, excellent photostability, and surface-bound HA make NPs a potential photothermal agent for tumor therapy.

### 3.3. PTT of Cancer Cells In Vitro

CCK-8 cell viability assay was used to evaluate the biocompatibility of HA-Au@SiO_2_@Au NPs at different NP concentrations (12.5, 25, 50, 100, 200, and 400 μg·mL^−1^). Clearly, no significant change in the cell viability of both PC-3 cells and MB49 cells was observed when NP concentration was less than 200 μg·mL^−1^ compared with the control group, indicating that HA-Au@SiO_2_@Au NPs have excellent biocompatibility (Figure 3A,D). This is likely contributed to the biocompatibility SiO_2_ shell and surface-modified HA. Even NP concentration reached 400 μg·mL^−1^, cell viability only decreased by 18% and 16%, suggesting negligible cytotoxicity even under such a high NP concentration.

Given the high photothermal conversion rate and low cytotoxicity induced by 100 μg·mL^−1^ HA-Au@SiO_2_@Au NPs, this concentration was chosen to photothermally ablate PC-3 cells and MB49 cells as a function of laser power density (0.5, 1, 1.5, and 2 W·cm^−2^) and irradiation time (100, 200, 300, 400, 600, 800, and 1000 s). The results revealed that cell viability decreased by more than 80% (84% and 85%) at 1.5 W·cm^−2^ for 600 s (Figure 3B,E). Similarly, the cell viability of both PC-3 cells and MB49 cells gradually decreased upon increasing the irradiation time (Figure 3C,F). In particular, the cell activity was decreased by more than 80% (84% and 86%) after 600 s irradiation at 1.5 W·cm^−2^. Thus, the combination of NP concentration of 100 μg·mL^−1^, laser power density of 1.5 W·cm^−2^, and irradiation time of 600 s is an optimal condition for PTT of cancer cells in vitro.

### 3.4. Fluorescence Imaging of Cancer Cells In Vitro

The fluorescence property of HA-Au@SiO_2_@Au NPs enables their use in cell imaging. PC-3 cells and MB49 cells overexpressing CD44 receptors were incubated with 100 μg·mL^−1^ Au@SiO_2_@Au NPs and HA-Au@SiO_2_@Au NPs, respectively. Changes in fluorescence intensity were compared at different incubation times (0.5, 1, 1.5, and 2 h). Compared with the treatment of Au@SiO_2_@Au NPs, the treatment of HA-Au@SiO_2_@Au NPs enabled both PC-3 cells (Figure 4A) and MB49 cells (Figure 4B) to display a higher fluorescence intensity, verifying the targeting capability of HA-Au@SiO_2_@Au NPs to CD44-overexpressed cancer cells.

### 3.5. PTT of Tumors In Vivo

Tumor-bearing mice were randomly divided into four groups (PBS, PBS + NIR, HA-Au@SiO_2_@Au, and HA-Au@SiO_2_@Au + NIR). PTT of tumors was performed at NP concentration of 100 μg·mL^−1^, laser power density of 1.5 W·cm^−2^, and irradiation time of 600 s. Tumor volume was measured every three days. The change curve of tumor volume with treatment time is shown in Figure 5. Our results revealed that the volume of prostate tumors (Figure 5A) and bladder tumors (Figure 5B) gradually decreased with the increased time in the HA-Au@SiO_2_@Au + NIR group. The prostate tumors and bladder tumors disappeared at 15 and 18 d posttreatment, respectively, with no recurrence. These results suggested that HA-Au@SiO_2_@Au NPs are a potential photothermal agent for elimination of PC-3 cells and MB49 cells in vivo.

### 3.6. Discussion

First, Au NPs with excellent fluorescence property were prepared using GSH as template, which were then coated with SiO_2_ shell via the sol-gel method. Next, Au nanoclusters were then deposited on the particle surface through seed growth method, followed by decoration of HA via EDC chemistry. The results revealed that the photothermal conversion efficiency of biocompatible HA-Au@SiO_2_@Au NPs was estimated to be 47.6%. In vitro photothermal experiment showed that more than 80% of cancer cells were photothermally ablated under the optimal PTT conditions (NP concentration of 100 μg·mL^−1^, laser power density of 1.5 W·cm^−2^, and irradiation time of 600 s). Notably, the specific accumulation of HA-modified NPs in tumor sites significantly improved PTT efficacy along with minimal systemic side effects. In the end, during the in vivo experiment, the tumor volume of mice after treated with HA-Au@SiO_2_@Au NPs reduced by laser irradiation, and the tumors disappeared at 15 and 18 d, respectively, with no sign of recurrence.

## 4. Conclusions

In this study, we prepared HA-Au@SiO_2_@Au NPs with the capabilities of CD44 receptor targeting, fluorescence imaging, and photothermal conversion for efficient PTT of tumors in vivo. TEM images and Zeta potential diagram of Au@SiO_2_@Au and HA-Au@SiO_2_@Au proved that CD44 molecules were successfully linked to the surface of Au@SiO_2_@Au NPs. The average size of HA-Au@SiO_2_@Au NPs was found to be about 37.5 nm and exhibit a high photothermal conversion efficiency (47.6%). At the NP concentrations (100 μg·mL^−1^), the HA-Au@SiO_2_@Au NPs with good photostability could show good photothermal effect and rapidly increased the temperature by approximately 40 °C (39 °C and 38.5 °C) within 200 s. The HA-Au@SiO_2_@Au NPs showed excellent biocompatibility at the effective NP concentrations (100 μg·mL^−1^). Thus, at the effective concentrations, the combination of laser power density of 1.5 W·cm^−2^, and irradiation time of 600 s was confirmed their potential applications in PTT of tumors in vitro. The targeting ability of HA makes NPs specifically accumulate in either bladder tumor tissues or prostate tumor tissues. Tumors finally disappeared at the about 18th day postinjection, which indicates that the HA-Au@SiO_2_@Au NPs can be used as an efficient photothermal agent for PTT of CD44 highly expressed tumors. In addition, both in vivo and in vitro experiments confirmed that HA-Au@SiO_2_@Au NPs hold great promise in uses as a photothermal agent for further translational medicine.

## Data Availability

Not applicable.

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
