# Peer review of "Hyaluronic Acid Modified Au@SiO2@Au Nanoparticles for Photothermal Therapy of Genitourinary Tumors"

_polymers, 2022, doi:10.3390/polym14214772_

Round 1
Reviewer 1 Report
The research article entitled “Hyaluronic Acid Modified Au@SiO2@Au Nanoparticles for Photothermal Therapy of Genitourinary Tumors” have investigated the photothermal ablation capability of HA-Au@SiO2@AuNPs in the invasive treatment of genitourinary tumors. The work reported in this manuscript is not well presented, with multiple serious technical flaws found regarding the fluorescent ability of Au nanoparticles (NPs), characterization of prepared Au nanoparticles, design of experiments, etc. The article has many grammatical and sentence errors, and the language organization needs to be improved. For these reasons, I conclude that the paper is not suitable for publication.
1. Authors are stating as fluorescent Au nanoparticles and fluorescence imaging in the abstract and conclusion. But results have been found for fluorescence of Au nanoparticles. Authors need to do photoluminescence spectroscopy of all the nanoparticles, HA-Au@SiO2@Au NPs, AuNPs as prepared, and Au@SiO2@Au.
2. Serious problem in confirmation of AuNPs formation. No characterization has been done
Authors need to do Uv-Vis spectroscopy, XRD, EDX, TEM, and FTIR for HA-Au@SiO2@Au NPs, AuNPs as prepared, and Au@SiO2@Au for the formation of AuNP alone as well as the formation of composite HA-Au@SiO2@Au NPs and Au@SiO2@Au.
3. FTIR and XRD comparative study of Au@SiO2@Au NPs, AuNPs as prepared, and Au@SiO2@Au with controls of HA and SiO2 alone have to be carried out.
4. For Fluorescence Imaging of Cancer Cells in Vitro, Cell culture images need to provide as proof.
5. For PTT of Tumors In Vivo study, no mice and tumor measurement images have been not provided. Simply graphs can not be efficient proof of the study.
6. The conclusion is very poor and needs to be improved
7. There are many grammatical and sentence errors in the article, and the language organization needs to be improved.
Author Response
Reviewer 1
- Authors are stating as fluorescent Au nanoparticles and fluorescence imaging in the abstract and conclusion. But results have been found for fluorescence of Au nanoparticles. Authors need to do photoluminescence spectroscopy of all the nanoparticles, HA-Au@SiO2@Au NPs, AuNPs as prepared, and Au@SiO2@Au.
Reply: We would like to thank the reviewer for the thoughtful comments. And it has been reported (ref 28-31) that the nanoparticles prepared by this method have fluorescence properties, and the similar work had been completed in our previous work (ref. 31).
- Serious problem in confirmation of AuNPs formation. No characterization has been done. Authors need to do Uv-Vis spectroscopy, XRD, EDX, TEM, and FTIR for HA-Au@SiO2@Au NPs, AuNPs as prepared, and Au@SiO2@Au for the formation of AuNP alone as well as the formation of composite HA-Au@SiO2@Au NPs and Au@SiO2@Au.
Reply: We respectfully agree incompletely with the reviewer's opinion. While there is some truth in this statement, the main materials involved in this study has been prepared by referring to the previous work (Reference 28-31), and tested strictly by Uv-Vis spectroscopy, XRD, EDX, TEM, etc. Obviously, we focus on the biological application of this material in this study, and the similar work had been completed in our previous work (ref. 31). So, the mentioned tests are repetitive and can be omitted to make the length of the article more reasonable.
- FTIR and XRD comparative study of Au@SiO2@Au NPs, AuNPs as prepared, and Au@SiO2@Au with controls of HA and SiO2 alone have to be carried out.
Reply: We respectfully thanks again for the reviewer's opinion. As mentioned above, we focus on the biological application in this study. So, the mentioned tests are repetitive and can be omitted. TEM image of the prepared HA-Au@SiO2@Au NPs is shown, but not very clear likely due to the coating of a large amount of HA on the particle surface as the previous work (Hu Y, et al. J Mater Chem B 2015, 3 (47), 9098-9108) reported. Zeta potential analysis revealed that Au@SiO2@Au and HA-Au@SiO2@Au NPs were negatively charged (-7.2 ± 3.8 and −18.8 ± 4.2 mV), proving that the NPs’ surface was modified with the modifier HA to allow for cell recognition.
- For Fluorescence Imaging of Cancer Cells in Vitro, Cell culture images need to provide as proof.
Reply: We appreciate the reviewer for the comments. Anyway, this assertion of “Fluorescence Imaging of Cancer Cells” is inaccuracy. We aim to analyze the interaction between the cells and materials, so the original text has been changed to “Quantitative Analysis of Cancer Cells Phagocytosis”. It can be found on page 7 highlighted in yellow.
- For PTT of Tumors In Vivo study, no mice and tumor measurement images have been not provided. Simply graphs can not be efficient proof of the study.
Reply: We regret greatly that the animals have been killed and the measurement pictures are unable to provide. Considering the time factor and the cause of the epidemic of covid-19, the animal experiment cannot be repeated again. Once again, we regret this greatly, and sincerely beg the reviewer’s understanding of our insurmountable difficulties.
- The conclusion is very poor and needs to be improved.
Reply: We are well aware of the the importance of the reviewer’s comment about the conclusion section and thank him/her so much. The necessary changes highlighted in yellow can be found in section “4. Conclusion” on page 13.
- There are many grammatical and sentence errors in the article, and the language organization needs to be improved.
Reply: Thanks for pointing out the language problems of this paper. We have accordingly made the necessary changes (highlighted in yellow) to the manuscript.
Reviewer 2 Report
The Authors have demonstrated the synthesis of Au@SiO2@Au Nanoparticles for Photothermal Therapy of Genitourinary Tumors. The work is interesting but requires more analysis before considering the manuscript.
1) Abstract: The abstract needs to be more quantitatively written.
2) Introduction: The introduction should cover the literature on the importance of silica/ gold with synthesis techniques of Gold/Silica and its various applications in recent times.
Here are some suggestions:
1) Schlather AE et al. Hot hole photoelectrochemistry on Au@ SiO2@ Au nanoparticles. The Journal of Physical Chemistry Letters. 2017 May 4;8(9):2060-7.
2) Utsav et al. Thermal crowning mechanism in gold–silica nanocomposites: plasmonic-photonic pairing in archetypal two-dimensional structures. Physical Chemistry Chemical Physics. 2021;23(32):17197-207.
3) Khanna S, et.al Fabrication of long-ranged close-packed monolayer of silica nanospheres by spin coating. Colloids and Surfaces A: Physicochemical and Engineering Aspects. 2018 Sep 20;553:520-7.
Result: The results and discussion section lack various characterizations for gold/silica/gold structure which must include FESE, Ramam, and XRD analysis to determine the morphological and structural part of the formed complex.
The silica formed was amorphous or crystalline, or porous? if porous, kindly add the BET analysis to it.
The result obtained should be compared with the previously reported literature, and a table can clearly depict the difference between the current complex compared to the literature.
Conclusion: The section is well written.
There are some grammatical errors that can be addressed in the revised version.
Author Response
Reviewer 2
1) Abstract: The abstract needs to be more quantitatively written.
Reply: Thanks for pointing out the problems of this paper in abstract. We have made the necessary changes (highlighted in yellow) to the manuscript.
2) Introduction: The introduction should cover the literature on the importance of silica/ gold with synthesis techniques of Gold/Silica and its various applications in recent times.
Here are some suggestions:
1) Schlather AE et al. Hot hole photoelectrochemistry on Au@ SiO2@ Au nanoparticles. The Journal of Physical Chemistry Letters. 2017 May 4;8(9):2060-7.
2) Utsav et al. Thermal crowning mechanism in gold–silica nanocomposites: plasmonic-photonic pairing in archetypal two-dimensional structures. Physical Chemistry Chemical Physics. 2021;23(32):17197-207.
3) Khanna S, et.al Fabrication of long-ranged close-packed monolayer of silica nanospheres by spin coating. Colloids and Surfaces A: Physicochemical and Engineering Aspects. 2018 Sep 20;553:520-7.
Reply: We are very grateful for the reviewer's constructive comments and insightful suggestions for improving the quality of this manuscript. So, the aforementioned literatures have been added, the corresponding changes (highlighted in yellow) can be found in section “1 Introduction” on page 4.
Result: The results and discussion section lack various characterizations for gold/silica/gold structure which must include FESE, Ramam, and XRD analysis to determine the morphological and structural part of the formed complex.
Reply: We respectfully agree partly with the reviewer's opinion. While there is some truth in this statement, the main materials involved in this study has been prepared by referring to the previous work (ref. 28-31), and tested strictly by Uv-Vis spectroscopy, XRD, EDX, TEM, etc. Obviously, we focuse on the biological application of this material in this study, and the similar work had been completed in our previous work (ref. 31). So, the mentioned tests are repetitive and can be omitted to make the length of the article more reasonable.
The silica formed was amorphous or crystalline, or porous? if porous, kindly add the BET analysis to it.
Reply: We greatly the reviewer for his/her helpful and professional comments. The silica formed was amorphous.
The result obtained should be compared with the previously reported literature, and a table can clearly depict the difference between the current complex compared to the literature.
Reply: Thanks a lot for the reviewer’s comments. As mentioned previously by the reviewer, the various applications of gold–silica nanocomposites are widepread in recent times. Compared to those materials, it is the only difference that hyaluronic acids (HA) were linked to the NPs in this study, and our focal point is the biological application of the material. So we think that the table can be omitted to make the length of the article more reasonable.
Conclusion: The section is well written.
Reply: We really appreciate the reviewer's affirmation.
There are some grammatical errors that can be addressed in the revised version.
Reply: Thanks for pointing out the language problems of this paper. We have accordingly made the necessary changes (highlighted in yellow) to the manuscript.
Reviewer 3 Report
The manuscript entitled “Hyaluronic Acid Modified Au@SiO2@Au Nanoparticles for Photothermal Therapy of Genitourinary Tumors” explains the preparation of hyaluronic acid modified Au@SiO2@Au nanoparticles and their effectiveness for the treatment of genitourinary tumors. Overall, the paper has been prepared well. However, the following points must be considered before further processing of the manuscript:
1- The final paragraph of the introduction explains the results of the study. It is not common to explain the results at the end of introduction. It is suggested just to mention the aim or final goal of investigation at the end of introduction. It is recommended to revise the paragraph (page 2, lines 64-76).
2- There are some abbreviations in Scheme 1 (e.g.; TEOS, HA, APTES,...). These abbreviations must be defined in figure caption.
3- The result and methods sections must be consistent. The characterization section in the results has not been described in the methods. There is no explanation for TEM imaging and zeta potential measurement in the methods.
4- The characterization of nanostructure is poor. It is not clear whether the claimed structures have been synthesized or not. For example, it is not clear if HA has been conjugated on the surface of Au@SiO2@Au NPs. It is necessary to prove that these structures have been prepared as they are expected. In the current form, it is not possible to accept these structures.
5- Why did not the authors measure the particle size using dynamic light scattering (DLS) method?
6- How did the authors prepare Au@SiO2@Au and HA-Au@SiO2@Au NPs for in vitro and in vivo applications? Did the authors prepare the NPs in water or a buffer system? Did they adjust pH and osmolarity before treatment of the cancer cells?
7- Why did the authors use the concentration of 100 μg·mL−1 HA-Au@SiO2@Au NPs for in vitro assay? Since there was no significant difference between different concentrations, why did not use the lowest concentration?
Author Response
Reviewer 3
1- The final paragraph of the introduction explains the results of the study. It is not common to explain the results at the end of introduction. It is suggested just to mention the aim or final goal of investigation at the end of introduction. It is recommended to revise the paragraph (page 2, lines 64-76).
Reply: Thanks for pointing out the problem of this section. They have been reduced and modified more rationally. The necessary changes (highlighted in yellow) to the manuscript can be found on page 5.
2- There are some abbreviations in Scheme 1 (e.g.; TEOS, HA, APTES,...). These abbreviations must be defined in figure caption.
Reply: Thanks a lot for the reviewer’s comments. The opinion is very important and we have adopted the proposal. The new caption can be found in the revised manuscript.
3- The result and methods sections must be consistent. The characterization section in the results has not been described in the methods. There is no explanation for TEM imaging and zeta potential measurement in the methods.
Reply: Many thanks to the reviewer for his/her helpful comments. The corresponding sentences were added into the revised manuscript. They (highlighted in yellow) can be found in section “2.1. Reagents and Instruments” on page 5.
4- The characterization of nanostructure is poor. It is not clear whether the claimed structures have been synthesized or not. For example, it is not clear if HA has been conjugated on the surface of Au@SiO2@Au NPs. It is necessary to prove that these structures have been prepared as they are expected. In the current form, it is not possible to accept these structures.
Reply: We are very grateful for the reviewer's constructive comments. As said in this study, we referred to previous reports and preparaed the materials, which had been fully characterized. And obviously, we focused on the biological application of this material in this study. So, the superfluous tests are repetitive and can be omitted to make the length of the article more reasonable. Zeta potential analysis revealed that Au@SiO2@Au and HA-Au@SiO2@Au NPs were negatively charged (-7.2 ± 3.8 and −18.8 ± 4.2 mV), fully proving that the NPs’ surface was modified with the modifier HA to allow for cell recognition.
5- Why did not the authors measure the particle size using dynamic light scattering (DLS) method?
Reply: Many thanks to the reviewers for this helpful comments. The particle size distribution of synthesized NPs in an aqueous colloidal solution is usually determined using the DLS technique. However, NPs that are determined by TEM are in the dried state, and the grain size directly reflects the size of particles. Therefore, TEM was applied instead of DLS in this study.
6- How did the authors prepare Au@SiO2@Au and HA-Au@SiO2@Au NPs for in vitro and in vivo applications? Did the authors prepare the NPs in water or a buffer system? Did they adjust pH and osmolarity before treatment of the cancer cells?
Reply: Many thanks to the reviewers for this important detail. PBS solutions of Au@SiO2@Au and HA-Au@SiO2@Au NPs were prepared for in vitro and in vivo applications. It was mentioned and highlighted in yellow in section “3.2. Photothermal Properties” on page 9 .
7- Why did the authors use the concentration of 100 μg·mL−1 HA-Au@SiO2@Au NPs for in vitro assay? Since there was no significant difference between different concentrations, why did not use the lowest concentration?
Reply: We greatly appreciate the reviewer for the comments. It needs to be explained detailedly. The maximum temperature of HA-Au@SiO2@Au NPs is positively correlated with the concentration at a given range, whereas the difference in maximum temperature was not obvious when the concentration was higher than 100 μg·mL−1. So we used the concentration of 100 μg·mL−1 HA-Au@SiO2@Au NPs for in vitro assay. It was mentioned and highlighted in yellow in section “3.2. Photothermal Properties” on page 9 .
Round 2
Reviewer 1 Report
Satisfied with the rebuttal for every query raised. The article is accepted for publication.
Reviewer 2 Report
The authors have addressed all the comments and have made significant modification to the manuscript. The manuscript can be acceptable in the current form.
Reviewer 3 Report
Dear Editor
Dear Authors
I believe the manuscript has NOT been sufficiently improved to warrant publication in Polymers.
Sincerely,